# Photodegradation of Lipofuscin in Suspension and in ARPE-19 Cells and the Similarity of Fluorescence of the Photodegradation Product with Oxidized Docosahexaenoate

**DOI:** 10.3390/ijms23020922

**Published:** 2022-01-15

**Authors:** Małgorzata B. Różanowska, Bartosz Różanowski

**Affiliations:** 1School of Optometry and Vision Sciences, Cardiff University, Cardiff CF24 4HQ, UK; 2Cardiff Institute for Tissue Engineering and Repair (CITER), Cardiff University, Cardiff CF10 3NB, UK; 3Institute of Biology, Pedagogical University of Kraków, 30-084 Kraków, Poland; Bartosz.Rozanowski@up.krakow.pl

**Keywords:** lipofuscin, retina, retinal pigment epithelium, docosahexaenoate, docosahexaenoic acid, fluorescence, photodegradation, photobleaching, cell viability, endocytic activity

## Abstract

Retinal lipofuscin accumulates with age in the retinal pigment epithelium (RPE), where its fluorescence properties are used to assess retinal health. It was observed that there is a decrease in lipofuscin fluorescence above the age of 75 years and in the early stages of age-related macular degeneration (AMD). The purpose of this study was to investigate the response of lipofuscin isolated from human RPE and lipofuscin-laden cells to visible light, and to determine whether an abundant component of lipofuscin, docosahexaenoate (DHA), can contribute to lipofuscin fluorescence upon oxidation. Exposure of lipofuscin to visible light leads to a decrease in its long-wavelength fluorescence at about 610 nm, with a concomitant increase in the short-wavelength fluorescence. The emission spectrum of photodegraded lipofuscin exhibits similarity with that of oxidized DHA. Exposure of lipofuscin-laden cells to light leads to a loss of lipofuscin granules from cells, while retaining cell viability. The spectral changes in fluorescence in lipofuscin-laden cells resemble those seen during photodegradation of isolated lipofuscin. Our results demonstrate that fluorescence emission spectra, together with quantitation of the intensity of long-wavelength fluorescence, can serve as a marker useful for lipofuscin quantification and for monitoring its oxidation, and hence useful for screening the retina for increased oxidative damage and early AMD-related changes.

## 1. Introduction

Retinal pigment epithelium (RPE) is a monolayer of neuroepithelial cells separating the retina from its choroidal blood supply. RPE cells gradually accumulate a product of incomplete lysosomal digestion of phagocytosed photoreceptor outer segments with age, as well as some autophagocytosed material known as age pigment or lipofuscin [1,2,3]. The greatest accumulation occurs in the macula, which is the area of the retina responsible for acute vision [2,4,5,6]. It has been determined that lipofuscin occupies 19% of the cytoplasmic volume of RPE cells from the macular area of 81–90-year-old cadavers [6]. Accelerated accumulation of lipofuscin is observed in some inherited retinal diseases such as Stargardt disease, fundus flavimaculatus, Batten disease (also known as neuronal ceroid lipofuscinosis) and bestrophinopathies [7,8,9].

When excited with light, RPE lipofuscin emits a characteristic golden-yellow fluorescence with a broad maximum at about 600 nm and large variations between individual granules (the maxima vary from 550 to 645 nm) [3,10,11,12,13]. It has been determined that, when excited with blue light, the fluorescence is emitted mostly by yellow-emitting fluorophores [12]. However, a considerable part of that emission originates not from a direct excitation of these fluorophores but via energy transfer from other blue-light-absorbing chromophores, most of which have still not been identified.

Lipofuscin fluorescence can be imaged in the retina in vivo using excitation in the range of 488 nm to 550 nm, and such imaging is used for monitoring retinal health and the progression of several retinal diseases [4,7,8,14,15,16,17,18,19].

Interestingly, it has been determined that lipofuscin fluorescence in the range of 650–750 nm, emitted when photoexcited with 550 nm light, increases in the retina in a linear fashion up to the age of 70 years, and then decreases more steeply than its prior increase, despite no changes occurring in the RPE cell density [18,20]. There are several pieces of evidence suggesting that the decrease in fluorescence in the retina could be due, at least in part, to photobleaching. In four out of five Stargardt disease patients monitored over a period of one year or longer, an overall reduction in lipofuscin fluorescence induced by excitation with 488 nm light intensity was reported, despite no cell loss occurring [21]. In this study, the reduction in fluorescence intensity was smaller in fellow eyes where patients’ corneas were covered during the daytime by contact lenses blocking above 90% of the visible light entering the eye. There are reports demonstrating that translocation of the re-attached neural retina after detachment reveals areas of increased autofluorescence intensity [22,23]. These hyperfluorescent areas visible after re-attachment match the shape of blood vessels, which, after the translocation, no longer screen the RPE behind them from incident light.

Moreover, it has been shown that lipofuscin fluorescence in the macaque retina in vivo decreases by about 20% as a result of acute exposure to a 210 J/cm^2^ light dose of 568 nm wavelength, while the integrity of the RPE monolayer remains intact [24,25,26]. In the case of human RPE ex vivo, a dose of 30.6 J/cm^2^ of 568 nm light was sufficient to cause fluorescence photobleaching [26].

Experiments, especially on albino mice with accumulated lipofuscin, also indicate that exposure to light can lead to a decrease in lipofuscin fluorescence [27,28]. A light-dependent fluorescence decrease was also observed in pigmented mice, both wild type and mice with deleted genes coding for the proteins responsible for clearance of all-*trans*-retinal, *abca4(-/-)rdh8(-/-)*, but the pigmented mice required longer breeding time for this to be detectable [29]. Mice that were raised under a light/dark cycle for 12 months exhibited a 2.8-fold decrease in RPE fluorescence in comparison to mice of the same genetic background raised in the dark.

The decrease in lipofuscin fluorescence is probably due to photo-oxidation of lipofuscin. We and others have shown that exposure of lipofuscin isolated from human RPE to light leads to the generation of reactive oxygen species and photo-oxidation of the lipofuscin itself [30,31,32,33,34,35,36,37,38,39,40].

It has been determined that the lipophilic extract from the RPE contains a number of different fluorophores [41,42], and several of them have been identified as various bisretinoids and their oxidation products [32,39,43,44,45,46,47,48,49,50,51,52]. These bisretinoids, except for the all-*trans*-retinal dimer, are formed as a result of the condensation of two molecules of vitamin A aldehyde (all-*trans*-retinal) with the ethanolamine group of phosphatidylethanolamine, and their subsequent transformations. They are considered to be the major emitters of yellow fluorescence from photoexcited RPE lipofuscin [12,43,44].

The effects of the photo-oxidation of A2E and other bisretinoids on their fluorescence properties have been studied both in solution and in cultured cells (reviewed in [43,53]), but no such studies have been performed on lipofuscin isolated from human RPE. Therefore, the aim of this study was to monitor the changes in fluorescence during the photo-oxidation of lipofuscin isolated from human RPE in suspension and in lipofuscin-laden ARPE-19 cells, as well as to determine whether photo-oxidation of intracellular lipofuscin can affect cell viability and endocytic function.

Lipofuscin contains abundant polyunsaturated fatty acids including docosahexaenoic acid (DHA) which, due to six unsaturated double bonds, is extremely susceptible to oxidation [48,54,55,56]. Oxidation products of DHA and other polyunsaturated lipids have been identified in the RPE lipofuscin [48,57,58,59]. Exposure of lipofuscin to light is likely to induce further oxidation of bisretinoids and DHA. We have shown that oxidation of DHA leads to the formation of products absorbing visible light with more potent photosensitizing properties than the lipophilic extract of lipofuscin, suggesting that even if they contribute only a small percentage to the light absorption of the lipofuscin, they could still be responsible for the majority of its photosensitizing properties [60]. Although the potential contribution of lipid oxidation products to the fluorescence has been studied previously, the study was limited to small end products of lipid oxidation such as malondialdehyde (MDA), 4-hydroxy-nonenal (HNE) and their adducts with amines, which upon absorption of ultraviolet light can emit blue fluorescence [61,62]. The absorption of visible light by oxidized DHA suggests the possibility that it can be excited by visible light and can contribute to lipofuscin fluorescence. Therefore, another aim of this study was to compare the fluorescent properties of oxidized DHA with those of lipofuscin and photodegraded lipofuscin.

To achieve these aims, we isolated lipofuscin from human RPE, exposed it to 9.76 mW/cm^2^ visible light and monitored its fluorescence after pre-selected exposure times of up to 10.5 h, using photoexcitation with wavelengths of 360 and 488 nm. DHA was auto-oxidized by exposure to the air, and its fluorescence properties were compared with lipofuscin before and after photodegradation. To determine whether photodegradation of lipofuscin can occur in cultured cells without causing cytotoxicity, confluent ARPE-19 cells were supplemented with lipofuscin added to the culture media. Cells with accumulated lipofuscin and control cells were exposed daily for 14 days to 45 min of 9.76 mW/cm^2^ visible light, so that the total radiant exposure of 369 J/cm^2^ was the same as for lipofuscin in suspension. Our results demonstrated that photodegradation of isolated RPE lipofuscin resulted in a decrease in the intensity of fluorescence above 550 and 500 nm when excited with 360 and 488 nm light, respectively, and led to an increase in lipofuscin fluorescence in the range of 380–530 nm observed with 360 nm excitation. A radiant exposure of 369 J/cm^2^ led to photodegraded lipofuscin with an emission spectrum with similarities to that of oxidized DHA. Exposure of lipofuscin-laden ARPE-19 cells to such a dose of visible light fractionated over 14 days resulted in removal of lipofuscin from cells, as well as a decrease in the long-wavelength fluorescence and an increase in the short-wavelength fluorescence. Importantly, the photodegradation of lipofuscin did not affect cell viability or endocytic activity. Interestingly, oxidized DHA contributed to the emission at about 600 nm, which is thought to be characteristic for bisretinoids. Taken together, our results demonstrated that photodegradation of lipofuscin could be responsible for the observed loss of golden-yellow fluorescence observed in clinical studies. It is suggested that the emission spectra of retinal lipofuscin could serve as a marker of oxidative damage to the RPE in vivo; however, a shorter excitation wavelength than 488 nm would be required.

## 2. Results

### 2.1. Exposure of RPE Lipofuscin to Visible Light Results in a Decrease in Long-Wavelength Fluorescence and an Increase in Short-Wavelength Fluorescence

Isolated lipofuscin emitted broad-band fluorescence when excited with 360 nm or 488 nm light, in agreement with previous reports [11,12,13] (Figure 1A,B). The spectra, corrected for changes in the detection sensitivity of the spectrofluorometer, showed that the emission maximum at about 610 nm was at the same wavelength as reported previously (Figure 1C).

Exposure of lipofuscin to 9.76 mW/cm^2^ visible light resulted in photobleaching of yellow/orange fluorescence and an increase in blue-green fluorescence (Figure 1). In the case of excitation with 360 nm light, a decrease in emission intensity can be seen in the range of 550 to 700 nm. In the case of excitation with 488 nm light, a decrease in emission intensity can be seen over the entire range of the emission spectra starting from 500 nm.

During lipofuscin photodegradation, the intensity of the short-wavelength emission induced by 360 nm light increased in the range of 380 to 540 nm (Figure 1A,D). After 10.5 h of lipofuscin photodegradation via radiant exposure of 369 J/cm^2^, the intensity of emission at 450 nm increased 3.7-fold, while, at the same time, the emission at 610 nm decreased by 43% and 58%, respectively, when using excitation at 360 and 488 nm (Figure 1D,E). Emission at 530 nm exhibited an initial increase, which was followed by plateauing or a decrease depending on whether the emission was induced by excitation with 360 or 488 nm light. Similar but less pronounced changes could be seen for emission at 550 nm. We were interested in fluorescence changes at that wavelength because it has been reported that the ratio of fluorescence intensities at 550 nm and 600 nm is increased in RPE cells from eyes affected by age-related macular degeneration (AMD) in comparison with RPE cells isolated from healthy eyes of a similar age (reported by Yakovleva and colleagues), where the fluorescence was induced by excitation with 488 nm light [63]. AMD retinas are exposed to greater levels of oxidative stress than age-matched healthy retinas, as evidenced by the detection of increased levels of easily chelatable iron and products of lipid oxidation [64,65]. Therefore, it can be expected that the increased oxidative stress in AMD retinas can cause oxidation, including oxidation of lipofuscin, and the observed changes in fluorescence. Our results demonstrated that during photodegradation of isolated lipofuscin, there was an increase in the fluorescence ratio at 550 and 610 nm (Figure 1F,G). Such an increase is even more pronounced for the emission ratio at 530 and 610 nm. The kinetics of the ratio increase appear to be different for excitation with 360 nm light than for excitation with 488 nm light. The ratios of emission intensities at 450 and 610 nm or 470 and 610 nm exhibit linear increases with increasing radiant exposure, and therefore appear to be a better indicator of lipofuscin degradation than the ratios of emission intensities at 550 nm or 530 nm and 610 nm (Figure 1F).

### 2.2. The Fluorescence of Oxidized DHA Exhibits Similar Spectral Characteristics to Fluorescence Appearing during Photodegradation of Lipofuscin

DHA is an abundant component of lipofuscin. Due to its six unsaturated double bonds, it is very susceptible to oxidation. It has been shown that both oxidized DHA and lipofuscin can exhibit photosensitized generation of singlet oxygen and free radicals, which can lead to further oxidation of DHA [60,66,67]. Oxidized DHA exhibits a broad absorption spectrum extending up to 600 nm. Considering that the quantum yields of singlet oxygen generation by oxidized DHA are 0.22 and 0.13 for photoexcitation with 355 nm and 425 nm light, respectively [60], this means that up to 78 and 87% of the absorbed photons can be utilized in other deactivation pathways such as fluorescence. Therefore, we investigated the possibility that photoexcitation of oxidized DHA can result in the emission of fluorescence. Oxidized DHA emitted broad-band fluorescence when excited with either 360 nm or 488 nm light (Figure 2). Interestingly, the emission spectrum obtained by photoexcitation of oxidized DHA with 360 nm light was similar to the green and longer wavelength portion of the emission spectrum of photodegraded lipofuscin (Figure 2A).

Oxidized DHA also emitted fluorescence when excited with 488 nm blue light (Figure 2B). The emission spectrum showed similar spectral characteristics to the emission spectrum of lipofuscin before or after photodegradation, indicating that oxidized DHA can contribute to the fluorescence emission with a maximum at 610 nm, which so far has been attributed to the fluorescence of bisretinoids [14,53,68].

### 2.3. Exposure of Lipofuscin-Laden ARPE-19 Cells to Light Results in Loss of Lipofuscin from Cells and Changes in Fluorescence

To enrich ARPE-19 cells with lipofuscin, lipofuscin was added to the cell culture medium and supplemented to confluent cells during each media change over a period of 32 days. The cells were imaged using phase-contrast microscopy with low light intensity, in order to monitor lipofuscin accumulation and any signs of cell loss such as rounding up or loss of confluency. Accumulation of lipofuscin granules was clearly visible in the cell monolayer (Figure 3).

Fluorescence microscopy with a long-pass emission filter and a colour camera showed a characteristic golden-yellow fluorescence in cells fed with lipofuscin when excited with blue light, which was absent in cells without such supplementation (Figure 3). Splitting images in ImageJ into RGB channels allowed the observation that the red fluorescence was more intense than the green fluorescence in the lipofuscin-laden cells.

Based on previous reports, it can be expected that the radiant exposure of 369 J/cm^2^ needed for the development of oxidized DHA-like fluorescence could be phototoxic to cells, especially if they contain lipofuscin [24,25,26,48,69,70,71,72,73,74,75]. Friedman and Kuwabara observed severe retinal damage in the photoreceptors and changes to the retinal pigment epithelium in rhesus monkeys exposed for 15 min to light from an indirect ophthalmoscope providing a retinal irradiance of 270 mW/cm^2^ and a dose of 243 J/cm^2^ [70]. The experiments of Morgan and colleagues on macaque monkeys demonstrated that there was a very small difference of just 18% between the threshold dose required to bleach fluorescence and the threshold dose causing a disruption of the RPE monolayer [24,25,26]. The threshold dose causing visible RPE photodamage in macaque monkeys using a 568 nm laser was determined as 247 J/cm^2^. Radiant exposures sufficient to cause photodamage to the retina were even smaller when shorter irradiation wavelengths were used [26,72]. Using photoexcitation of macaque retinas in vivo with 488 nm light, photobleaching of fluorescence with some disruption of the RPE monolayer was observed after a radiant exposure of just 79 J/cm^2^, and the value was even smaller for exposure to 460 nm light [26,72]. Cultured RPE cells are also susceptible to photodamage, and it has been shown that internalized lipofuscin makes them even more susceptible to the phototoxic effects of light in comparison to cells without lipofuscin [48,59,71,73,74,75]. For this reason, we decided to fractionate the radiant exposure of 369 J/cm^2^ into 14 doses of 26.3 J/cm^2^, in order to make sure that the individual doses were not likely to cause detectable damage, and to give cells time to repair any sublethal damage before the next dose was applied. Hence, the cells were exposed daily for 45 min to a dose of 26.3 J/cm^2^ visible light, giving them 23 h each time for repair. The control plates with lipofuscin-laden and lipofuscin-free cells were incubated with PBS for 45 min in the dark.

The accumulated lipofuscin granule density and the fluorescence were stable over the period of 14 days after the supplementation ended and the cells were subjected to a daily media change and 45 min incubations with PBS in the dark (Figure 3). When the fluorescence images of lipofuscin-laden cells were split into green and red channels, the intensity of the red fluorescence appeared much greater than that of the green fluorescence, as in the cells which had just completed the period of supplementation with lipofuscin (Figure 3). However, when lipofuscin-laden cells were exposed daily to 26.3 J/cm^2^ visible light, both the granule density and fluorescence intensity decreased (Figure 3). When inspected via microscopy a day after the first exposure, there was clearly visible lipofuscin packeted into spherical structures floating above the cell monolayer. The monolayer appeared confluent, with no sign of cell rounding and detachment. After two exposures, the fluorescence of the packeted lipofuscin and lipofuscin in the cells appeared green. When the images were split into green and red channels, the intensity of the red fluorescence appeared similar to that of the green fluorescence (Figure 3). After completing all 14 exposures, the fluorescence intensity in both green and red channels, as well as the lipofuscin granule density, were considerably decreased in comparison with lipofuscin-laden cells maintained for 14 days in the dark (Figure 3). Cells without lipofuscin which were exposed to light exhibited a small increase in green fluorescence in comparison with cells maintained in the dark, but no granules could be seen in the cell monolayer (Figure 3).

It has been reported that antioxidants can inhibit the photo-oxidation of bisretinoids and ameliorate the phototoxic effects of lipofuscin [68,73]. To determine whether supplementation of cells with antioxidants during exposure to light can affect lipofuscin density and fluorescence, cells were exposed in the presence and absence of N-acetyl cysteine (NAC) or lipoic acid. Both of these compounds are powerful antioxidants [76]. NAC can scavenge several reactive oxygen species, as well as serving as a precursor of glutathione. Lipoic acid can also play an antioxidant role by scavenging reactive oxygen species and by binding redox-active metal ions such as iron or copper. It can also reduce, and thereby regenerate, the original protective form of other powerful antioxidants such as glutathione, vitamin E and vitamin C. The presence of NAC and lipoic acid during lipofuscin exposure to light did not affect lipofuscin granule density or fluorescence in comparison with cells that were not supplemented (Figure 4).

The quantification of the lipofuscin granule density and the intensity of fluorescence in the green and red channels revealed that the exposure to light resulted in a 6.6-fold decrease in granule density (Figure 5). Both green and red fluorescence decreased 3.6- and 5.3-fold, respectively (Figure 5). The ratio of fluorescence in the red channel to that in the green channel also decreased from 1.37 for cells incubated in the dark to 0.94 for cells exposed to 369 J/cm^2^ visible light.

While the colour changes in the emission due to exposure of the lipofuscin-laden cells to light are visible to the eye equipped with a fluorescence microscope and can be recorded with a colour camera, spectrofluorimetry enables the spectral changes in fluorescence to be investigated in more detail (Figure 6). Spectrofluorimetry of lipofuscin-laden cells maintained in the dark and solubilized for measurements with Triton X-100 revealed that the fluorescence emission spectra resembled the spectra recorded for lipofuscin alone but after 2–3 h of photodegradation (Figure 1 and Figure 6). The ratios of the fluorescence emission from lipofuscin-laden cells at 450/610 and 470/610 nm were 5.3 and 4.9, respectively. These values are in between the corresponding values for lipofuscin exposed to 70.3 and 105.4 J/cm^2^ light (Figure 1E). This spectral change in fluorescence occurred even though during the supplementation period the cells were kept in the dark and the media changes were performed under dim light. It has been reported previously that phagocytosis by RPE cells is associated with the generation of reactive oxygen species and lipid peroxidation [77]. Therefore, it appears plausible that such reactive oxygen species could lead to partial oxidation of lipofuscin, resulting in changes in its composition and, consequently, in its fluorescence properties. It should be noted that cells without lipofuscin maintained in the dark also exhibited fluorescence, particularly in the blue-green range, even though no granule accumulation could be seen in the cell monolayer. This could be due to oxidative modifications of intracellular molecules and/or the formation of lipofuscin-like material due to autophagy, with the size of the nondigestible remnants of autophagolysosomes not big enough to be visible with the microscope used. It has been reported previously that lipofuscin-like material can accumulate in cells during long-term culture as a result of autophagy of intracellular organelles [1].

The initial 530-to-610-nm intensity ratios of the fluorescence induced by 360 or 488 nm light in lipofuscin-laden cells were 3.7 and 0.45, respectively (Figure 6A,C). These values are similar to the corresponding ratios for lipofuscin photodegraded for about 90 and 10 min, respectively (Figure 1C,G). Such a discrepancy is not unexpected, considering that different excitation wavelengths were used. Lipofuscin is a complex mixture of various chromophores with different photosensitizing and fluorescent properties, and it can be expected that their susceptibility to oxidation may vary. It can be also expected that the spectral changes in lipofuscin fluorescence upon phagocytosis-related oxidation of lipofuscin may be different from those observed upon photo-oxidation, because lipofuscin photo-oxidation involves the photosensitized production of singlet oxygen whereas the phagocytosis process is accompanied by free-radical generation [60,66,67,77].

Comparison of the fluorescence emission induced by 360 nm light between lipofuscin-laden cells maintained in the dark and cells exposed to a fractionated dose of 369 J/cm^2^ visible light revealed a decrease in fluorescence for all emission wavelengths above 455 nm and an increase in fluorescence below that wavelength (Figure 6A). This marked increase in short-wavelength fluorescence was clearly visible, despite the fact that cells exposed to light lost 85% of the lipofuscin granules (Figure 5A and Figure 6A). It should be noted that cells without lipofuscin supplementation subjected to daily exposure to visible light exhibited an increase in fluorescence emission in the range of 420 to 570 nm when photoexcited with 360 nm light, compared with cells maintained in the dark (Figure 6A). These cells, however, showed no significant differences in fluorescence when photoexcited with 488 nm light (Figure 6B,C).

The intensity of 360 nm light-induced fluorescence emission at 610 nm from lipofuscin-laden cells decreased 2.6-fold as a result of exposure to light (Figure 6A). This value is greater than the 1.7-fold decrease observed in lipofuscin suspension exposed to the same dose of light (Figure 1A,C). This is consistent with a 6.6-fold loss of lipofuscin granules from cells, accompanied by the formation in the remaining lipofuscin of species photoexcitable by 360 nm light that can contribute, directly or indirectly, to fluorescence emission at 610 nm.

Exposure to light led to a decrease in fluorescence induced by 488 nm light over the whole range of emission wavelengths (Figure 6B,C). The intensity of 488 nm light-induced fluorescence emission at 610 nm from lipofuscin-laden cells decreased 5.1-fold as a result of exposure to light and became very similar to the fluorescence emitted by cells not supplemented with lipofuscin maintained in the dark or exposed to light (Figure 6B,C). The expected value of the decrease in 610 nm emission intensity can be calculated based on: (i) a 6.6-fold reduction in the density of lipofuscin granules, (ii) a 2.4-fold reduction in lipofuscin fluorescence at that wavelength observed as a result of exposure to 3.69 J/cm^2^ light, and (iii) an assumption that the lipofuscin isolated from RPE cells is the only contributor to fluorescence from ARPE-19 cells. This value (3.6-fold decrease) is considerable smaller than the observed 5.1-fold decrease, suggesting that oxidative changes in the lipofuscin granules due to phagocytosis and exposure to light were greater than those occurring in isolated lipofuscin exposed to light.

Supplementation of cells with N-acetyl cysteine or lipoic acid during exposure to light exerted only a small effect on the changes in the fluorescence of lipofuscin-laden cells induced by these exposures (Figure 6). The emission of fluorescence induced by excitation with 360 nm light was slightly smaller in the presence of these antioxidants than in their absence. The differences in fluorescence induced by 488 nm light between cells without antioxidants and cells supplemented with NAC or lipoic acid were even less pronounced.

### 2.4. Effect of Lipofuscin Photodegradation on Cell Viability and Endocytic Activity

The phototoxicity of lipofuscin has been reported previously, and therefore our exposure conditions were designed with the aim of avoiding lethal effects on cells [48,71,73,74,75]. The photo-oxidation of lipofuscin is likely to release reactive oxygen species, as well as polar products of oxidation of bisretinoids and polyunsaturated fatty acids, which can exert toxic effects unless detoxified and removed from the cell. Although bisretinoids such as A2E are strongly anchored within the lipofuscin granules [48] and therefore unlikely to diffuse out of the granule to exert their toxic effects, some their scission products resulting from oxidative degradation [49,52] are likely to be small polar molecules similar to the oxidation products of all-*trans*-retinal, which were shown to be highly (photo)toxic [78].

Therefore, we fractionated the dose of light to enable cells to detoxify and/or remove the toxic products and monitored the cell morphology and integrity of the monolayer via phase-contrast microscopy, which showed no indication of toxic effects of exposure to lipofuscin and/or light (Figure 3 and Figure 4). The MTT assay of the metabolic activity of cells performed 24 h after the final 14th exposure confirmed that there were no significant differences in viability between cells with and without lipofuscin, kept in the dark or exposed to light (Figure 7A).

One of the essential roles of RPE cells is endocytosis [79]. It has been shown in previous studies that internalized lipofuscin and/or oxidative stress can inhibit phagocytosis of the photoreceptor outer segments by cultured RPE cells [73,80,81,82,83,84,85,86]. To test whether accumulated lipofuscin and/or exposure to light can affect endocytosis, we used the neutral red assay to assess the endocytic activity of cells (Figure 7B). The assay was performed 23 h after the final (14th) exposure to light, and it demonstrated that there were no significant differences in endocytic activities between cells with and without lipofuscin, kept in the dark or exposed to light.

## 3. Discussion

### 3.1. Changes in the Long-Wavelength Fluorescence Emission during Photodegradation of Lipofuscin

Our results indicate that exposure of lipofuscin to visible light, both in suspension and in ARPE-19 cells, results in a decrease in fluorescence at about 600 nm, which is thought to be characteristic of RPE lipofuscin and is ascribed to the fluorescence of bisretinoids [53].

The monotonic decrease in fluorescence at 600 nm observed during irradiation of lipofuscin granules in suspension in PBS (Figure 1) may seem to contradict the claim that photo-oxidation of the lipofuscin components A2E and all-*trans*-retinal dimer is accompanied by a fluorescence increase when compared to the parent compound [47,53,68]. In the experiment descriptions reported by Kim and colleagues, the irradiation wavelength and the irradiation times were provided but the values of the irradiance or the radiant exposure were not provided [47,53,68], and therefore it is not possible to compare these with the irradiance or radiant exposures used in our experiments. There is a possibility that such an increase could have occurred before our first measurement of irradiated lipofuscin after delivering a radiant exposure of 17.6 J/cm^2^. Nevertheless, the results based on which such a conclusion about the fluorescence increase was drawn [47,68] are not sufficient to support it. 

In the first study [47], A2E was oxidized by exposure to endoperoxide of 1,4-dimethylnaphthalene, and then the mixture was subjected to HPLC with absorption detection at 430 nm, while the fluorescence induced by excitation with 430 nm light was monitored at 600 nm. The chromatograms of the mixture included evidence of A2E, isoA2E, monoperoxy-A2E and bisperoxy-A2E, as well as a number of other non-identified peaks. The ratios of fluorescence to absorption peaks in these chromatograms were greater for monoperoxy-A2E and bisperoxy-A2E than for A2E or isoA2E. This suggests that the quantum yield of emission at that particular wavelength of 600 nm with 430 nm excitation was greater for monoperoxy-A2E and bisperoxy-A2E than for A2E or isoA2E. However, this observation is not enough to enable the conclusion that the fluorescence of monoperoxy-A2E and bisperoxy-A2E is greater than that of A2E. A meaningful comparison should be based on comparing the whole emission spectra obtained by excitation of equimolar concentrations of A2E and known oxidation products, or by excitation of A2E and the whole mixture of oxidation products derived from that A2E. The same reasoning can be applied to other photo-oxidation products of A2E, namely, monofurano-A2E, bisfurano-A2E and all-*trans*-retinal dimer, for which it was also claimed that their fluorescence exceeded that of their parent compounds.

The second study that reported an increase in fluorescence intensity upon photo-oxidation of a bisretinoid [68] employed alkenyl ether lysoA2-phosphatidylethanolamine containing one plasmalogen with a fully saturated 18-carbon chain (LysoA2-PE(P18:0/0:0)). This bisretinoid was solubilized in 2% DMSO in phosphate buffer, and its fluorescence spectra were acquired using excitation with light at 440 nm during continuous exposure to that light. The initial emission maximum of LysoA2-PE(P18:0/0:0) was at 608 nm. Upon photo-oxidation, the emission intensity appeared to increase, while its maximum shifted towards the shorter wavelength of 594 nm. It should be considered that LysoA2-PE(P18:0/0:0) is hydrophobic and therefore likely to have low solubility in aqueous solution and to precipitate when solubilized in PBS with only 2% DMSO. It can be suggested that, as a result of photo-oxidation, more polar molecules are generated which, due to the increased polarity, are more soluble in the aqueous solvent. Support for such a scenario comes from experiments on A2E which, due to lack of a highly hydrophobic plasmalogen, is less hydrophobic than LysoA2-PE(P18:0/0:0) but still appears to aggregate when solubilized in PBS with 2% DMSO [87]. It was reported that A2E solubilized in 2% DMSO in PBS can be photoexcited not only at 436 and 480 nm but also at 545 nm wavelength, and it emits fluorescence with maxima at 620, 630 and 630 nm, respectively. However, A2E does not absorb 545 nm light. Therefore, it can be suggested that A2E forms aggregates in PBS with 2% DMSO, affecting its absorption and fluorescence properties.

In other studies from the same group, it was reported that exposure of A2E to 480 nm light resulted in a decrease in the intensity of fluorescence emission and a shift of the maximum from 612 nm to 586 nm [87,88]. Similar results were observed when A2E was photodegraded by 440 nm light and fluorescence was excited with 450 nm light [89]. These results are consistent with the observed fluorescence decrease in our study (Figure 1).

### 3.2. Decrease in Long-Wavelength Fluorescence during Photo-Oxidation of Lipofuscin and an Increase in Blue/Cyan or Green-to-Orange Fluorescence Ratio as a Marker of Lipofuscin Oxidation

Our results confirm the previous studies [53,68,90] and indicate that, due to spectral changes occurring in lipofuscin due to oxidation, the intensity of fluorescence alone is not sufficient for lipofuscin quantification. However, in combination with information from the fluorescence emission spectra, it can provide valuable information about the quantity of lipofuscin, as well as its oxidation state.

While the photo-oxidation of lipofuscin resulted in a monotonic decrease in fluorescence at 600 nm when excited with 488 nm light, the fluorescence emission at about 530 nm exhibited an initial increase up to a dose of about 50 J/cm^2^, which was then followed by a monotonic decrease with increasing radiant exposure (Figure 1E). This effect could be due to the initial oxidation of bisretinoids or other compounds, resulting in the formation of oxidation products emitting fluorescence at about 530 nm. Further oxidation of these products could lead to their photodegradation, accompanied by a further fluorescence decrease, as suggested in recent reviews [53,68,90].

Our results on the photo-oxidation of isolated RPE lipofuscin (Figure 1F) are consistent with the results of a comparison between the fluorescence emission spectra recorded in suspension of human RPE cells isolated from 74-year-old eyes affected with AMD and those isolated from 75-year-old eyes without any signs of AMD [63,91]. The fluorescence spectra from the AMD-affected RPE cells with excitation with 488 nm light show that the ratio of emission at 550 nm to that at 600 nm is increased compared with normal RPE cells of a similar age. Such an increase in the ratio can be seen during photo-oxidation of lipofuscin in suspension and in cultured cells, suggesting that it can be used as a marker of oxidative damage to the lipofuscin and/or RPE. However, due to the complex kinetics of the increases in the ratios of 550/610 nm or 530/610 nm intensities, monitoring that ratio for 450/610 or 470/610 nm intensities, where there is a linear increase during photodegradation, can be more straightforward to interpret. The disadvantage of the latter procedure is that it would require an excitation wavelength shorter than 488 nm, which is the wavelength currently used in clinical instruments. The shorter excitation wavelength could also impose an increased risk of phototoxicity (reviewed in [66]). Following the discovery that the phototoxicity thresholds of radiant exposures used for light safety standards were incorrect for 568 nm light [24,25,26], Zhang and colleagues determined the phototoxicity thresholds for several other wavelengths, including 460 nm light [72]. Further experiments are needed to determine whether using the excitation of lipofuscin and retinas with light of 460 nm wavelength or shorter can produce useful information about the lipofuscin content and its oxidation state, and whether it is feasible to use this approach in a clinical setting without risking retinal photodamage.

### 3.3. Oxidized DHA May Contribute to Lipofuscin Fluorescence

Our results clearly demonstrate that oxidized DHA can emit fluorescence when excited with 360 or 488 nm light (Figure 2). The spectral characteristics of that fluorescence indicate that it can contribute to the blue-green fluorescence of lipofuscin, particularly when lipofuscin is photo-oxidized (Figure 1 and Figure 2). This, together with several other pieces of evidence, argues that oxidized DHA can contribute to lipofuscin fluorescence since: (i) DHA accounts for 6–9% and 7–16% of fatty acids in lipofuscin phospholipids and free fatty acids, respectively [54]; (ii) upon exposure to light lipofuscin generates reactive oxygen species such as singlet oxygen, superoxide, hydroxyl radical and hydrogen peroxide, which can oxidize DHA [66,67]; (iii) an oxidation product of DHA, carboxyethylpyrrole, was identified in human RPE lipofuscin [48]; (iv) oxidized DHA generates, with high quantum yields, similar reactive oxygen species as lipofuscin [60]. The contribution of oxidized lipids to the lipofuscin fluorescence was considered in the past but dismissed based on studies of the fluorescent properties of the end products of lipid oxidation such as MDA and HNE, and their adducts with amines [10]. These particular products of lipid oxidation are very small molecules, which do not absorb visible light, and therefore UV light is required for their photoexcitation, upon which they emit blue fluorescence. It can be suggested that the products of oxidation of other lipids with fewer unsaturated double bonds, such as docosapentaenoic acid, arachidonic acid and linolenic acid, as well as small scission products of bisretinoid degradation, could contribute to the short-wavelength emission spectrum of lipofuscin, but this suggestion requires experimental verification.

### 3.4. Physiological Relevance

It is important to consider the physiological relevance of our study. The irradiance levels of 9.76 mW/cm^2^ employed in our study are about 100 times greater than the highest estimates of those encountered in daylight, which is 0.1 mW/cm^2^ for sunlight reflected from snow [92]. However, the cultured cells were exposed to that irradiance for only 45 min per day for 14 days, providing individual radiant exposures of 26.4 J/cm^2^ and a total radiant exposure of 369 J/cm^2^. Such a dose can be expected to accumulate in the human retina over a period of 1025 h of exposure to 0.1 mW/cm^2^, or a period ten-fold longer for 0.01 mW/cm^2^, which is considered to be the irradiance of the retina under typical lighting conditions [92]. Assuming further that the exposure to daylight is for 16 h per day, this gives 640 days over which such a dose could accumulate. 

An interesting observation from our experiments is the loss of lipofuscin granules from cultured cells as a result of exposure to light. While ARPE-19 cells accumulate large amounts of lipofuscin, which remains there for the following 15 days when cells are maintained in the dark, the exposure of lipofuscin-laden cells to light results in a substantial loss of lipofuscin granules, which appear as lipofuscin packeted into membrane-enclosed structures floating in the culture medium above the cell monolayer. Under experimental conditions in vitro, this packeted lipofuscin is removed when the culture medium is replaced with PBS before the next exposure to light. However, in the retina, such packeted lipofuscin may contribute to the age-related deposits accumulating between the RPE and the collagenous layer of Bruch’s membrane, which separates the retina from the choroidal blood supply. This is consistent with the observations of histologic sections of human retina of Burns and Feeney-Burns, where they observed membrane-enclosed parts of RPE cells containing lipofuscin budding of the cells and forming a part of deposits between the RPE and Bruch’s membrane, known as drusen [93]. Similar observations were made by Gouras and colleagues, who studied the retina of aged monkeys [94]. More recently, it has been reported that during the early stages of AMD there is a formation of lipofuscin-dense structures within RPE cells and a loss of lipofuscin granules from RPE [95]. It appears from our results that oxidative damage to the lipofuscin can trigger such effects. This can also explain why there is a loss of lipofuscin granules, and therefore a loss of fluorescence, in the aged retina and even more in AMD, where oxidative stress and oxidative damage is increased, as evidenced by the increased levels of easily chelatable iron and products of lipid oxidation, respectively [48,59,96,97]. Further investigations are clearly needed to understand the mechanism triggering the externalization of lipofuscin.

Our results showing the loss of lipofuscin granules from ARPE-19 cells and the spectral changes in fluorescence as a result of exposure to light may also help to explain why there an age-related decrease in lipofuscin fluorescence in the human retina and a light-induced decrease in the lipofuscin fluorescence in human and monkey retinas, and why there may be no spatial correlation between retinal fluorescence and bisretinoids, as observed in human and monkey retinas [98,99,100,101,102,103].

## 4. Materials and Methods

### 4.1. Reagents

Chemicals, of at least analytical grade, including lipoic acid, N-acetyl cysteine (NAC), dimethyl sulfoxide (DMSO), 3-(4,5-dimethylthiazol-2-yl)-2,5-diphenyltetrazolium bromide (MTT) and neutral red, were purchased from Sigma-Aldrich (Sigma-Aldrich, St. Louis, MO, USA) or Fisher Scientific (Loughborough, UK) unless stated otherwise. Phospholipid containing docosahexaenoate (DHA) acyl chains, 1,2-di-(4Z,7Z,10Z,13Z,16Z,19Z-docosahexaenoyl)-sn-glycero-3-phosphocholine (Di22:6PC) was from Avanti Polar Lipids (Alabaster, AL, USA). Chromatography-grade organic solvents were from Fisher, Merck or VWR International, and were used as supplied.

### 4.2. Isolation and Purification of RPE Lipofuscin

Research on human tissue was approved by the School Research Ethics Audit Committee, School of Optometry and Vision Sciences, Cardiff University. Human eyes were obtained from the Bristol Eye Bank, Bristol, UK. The research adhered to the tenets of the Declaration of Helsinki. Lipofuscin was isolated and purified from RPE cells from 74 pairs of human cadaver eyes (range of ages: 43–95 years, mean of 74 years), as described previously [38,48,60].

### 4.3. Preparation of Liposomes and Oxidation of DHA

Multilamellar lipid vesicles (liposomes) from Di22:6PC were prepared as described previously [38,60]. The lipid film was exposed to the air to allow for its oxidation and then hydrated with phosphate-buffered saline (PBS) to form liposomes. 

### 4.4. Absorption and Fluorescence Spectrometry

The optical density spectra of liposomal or lipofuscin suspensions were measured in a quartz cuvette with a 1 cm optical path length using a U-1800 UV–Vis spectrophotometer (Hitachi, Yamaguchi, Japan) equipped with Hitachi UV Solutions software. Fluorescence spectra were measured using a F4500 fluorescence spectrophotometer (Hitachi, Yamaguchi, Japan) equipped with Hitachi FL Solutions software. Fluorescein and rhodamine B were used to determine the correction of the spectral sensitivity of the spectrofluorometer detection.

### 4.5. ARPE-19 Cell Culture

ARPE-19 cells were purchased from the American Type Culture Collection (ATCC, Manassas, VA, USA) and cultured as described previously [48,78,104], using 24-well plates and DMEM/F12 with the addition of L-glutamine and penicillin–streptomycin, and 10% or 2% of heat-inactivated foetal calf serum (FCS) (Sigma-Aldrich Chemical Co., St Louis, MO, USA). All experiments were performed on confluent cell monolayers seeded in 24-well plates, with passage numbers 24 and 25, with media changes performed under dim room light (with the fluorescent light under the class II cabinet switched off).

### 4.6. Enriching ARPE-19 Cells with Lipofuscin

To enrich cells with lipofuscin, cells were fed with 1 mL of culture medium per well supplemented with 2.9 × 10^8^ lipofuscin granules per ml, as described previously [48]. Cells were fed with lipofuscin three times per week up to 13 times. Control cells were fed with culture medium supplemented with PBS as the vehicle. Incorporation of lipofuscin was monitored by bright-field/phase-contrast microscopy using dim light to avoid unnecessary photo-oxidation due to imaging [105].

### 4.7. Irradiation of Lipofuscin and ARPE-19 Cells with Visible Light

Suspensions of lipofuscin in PBS or cultured ARPE-19 cells were irradiated with visible light in spectrofluorometric cuvettes and in 24-well cell culture plates, respectively. A solar simulator Sol lamp (Honle UV Ltd., Birmingham, UK) was used as a light source, as described previously [48,78]. Filters absorbing heat and the residual UV light (Lee Heat Shield and #226 Lee UV filter, respectively; Lee Filters, UK; and a UV cut-off filter, AMO Inc., Santa Ana, CA, USA; 2 g/100 mL solution of copper sulphate) were placed between the lamp and the glass plate where the cuvette or cell culture plate was positioned. Fluence rates and irradiance spectra were measured using a spectroradiometer (Specbos 1201 with JETI LiMeS software; Glen Spectra, Stanmore, UK) (Figure 8). The irradiance used in the experiments was 9.76 mW/cm^2^, with an illuminance of 35.3 klx.

ARPE-19 cells were irradiated daily for 14 days, for 45 min each time, at a temperature of about 26 °C. Before exposure, the medium was removed and cells were washed twice with Dulbecco’s phosphate-buffered saline with calcium and magnesium (PBS), and for the exposure time, PBS with or without antioxidants (0.2 mM N-acetyl cysteine, NAC or 0.1 mM lipoic acid, LPA with 0.2% DMSO) was added to the wells. After exposure, PBS was replaced with a culture medium containing 2% FCS, and cells were returned to the incubator for 23 h before the next exposure or assessments of fluorescence, cell viability or endocytic activity assays were performed. Dark-maintained control cells were treated the same way, except the exposure to light was replaced by incubation of plates wrapped in blackened aluminium foil in the same room as the light source.

### 4.8. Quantification of Pigment Granules in Cell Monolayers and Their Fluorescence

Cells were visualized using an Olympus IX70 inverted microscope with phase contrast and fluorescence [78]. Fluorescence was induced by excitation with blue light; the emission was collected after passing through a long-pass emission filter. Images were obtained using a SPOT RT colour CCD camera and SPOT Advanced software (Diagnostic Instruments Inc, Sterling Heights, MI, USA).

ImageJ was used for quantification of the lipofuscin granule density and their fluorescence. In the bright-field and phase-contrast images, lipofuscin granules were visible as dark dots. They were quantified as an area above the threshold, which was determined using images of cells not supplemented with lipofuscin and expressed as percentage of the total image area. Fluorescence images were opened in ImageJ as an RGB stack, which splits them into three colour channels. Fluorescence in the green and red channels was quantified as total fluorescence intensity per image in each channel separately.

### 4.9. Monitoring of ARPE-19 Cells Viability and Endocytic Activity

Cell viability was assessed by monitoring the morphology of the ARPE-19 cell monolayer via phase-contrast microscopy using an Olympus IX70 inverted microscope and by MTT assay of reductive activity where cells were exposed for 1 h to cell culture medium containing 0.5 mg/mL of MTT, as described previously [78,104]. Then, cells were washed with PBS, MTT-derived formazan was solubilized using acidified isopropanol and absorbance at 570 nm was read using a Multiskan Ascent plate reader (Thermo LabSystems, Vantaa, Finland). Endocytic activity was determined by neutral red assay, where cells were exposed for 1 h to cell culture medium containing 0.1 mg/mL of neutral red followed by extraction and absorbance reading at 545 nm [106].

## 5. Conclusions

In conclusion, our results demonstrated that exposure of lipofuscin to physiologically relevant doses of visible light led to a decrease in its long-wavelength fluorescence at about 610 nm, with a concomitant increase in the short-wavelength fluorescence in the blue-cyan spectral range. The emission spectrum of photodegraded lipofuscin exhibited similarity with that of oxidized DHA. Oxidized DHA contributed to the emission at about 600 nm, thought to be characteristic of the bisretinoid components of lipofuscin. Exposure to light of lipofuscin-laden cells led to a loss of lipofuscin granules from the cells, while retaining cell viability and endocytic activity. The spectral changes in fluorescence in lipofuscin-laden cells resembled those seen during photodegradation of isolated lipofuscin. Our results demonstrated that photodegradation of lipofuscin may be responsible for the observed loss of golden-yellow fluorescence observed in clinical studies, and that the fluorescence emission spectra, together with quantitation of the intensity of long-wavelength fluorescence, could serve as a marker useful for lipofuscin quantification and for monitoring its oxidation, and hence also useful for screening the retina for increased oxidative damage and early AMD-related changes. However, excitation with a wavelength shorter than 488 nm would be required, and therefore further research is needed to establish the optimal excitation wavelength which would allow lipofuscin photo-oxidation to be assessed while remaining safe for the retina.

## Figures and Tables

**Figure 1 ijms-23-00922-f001:**
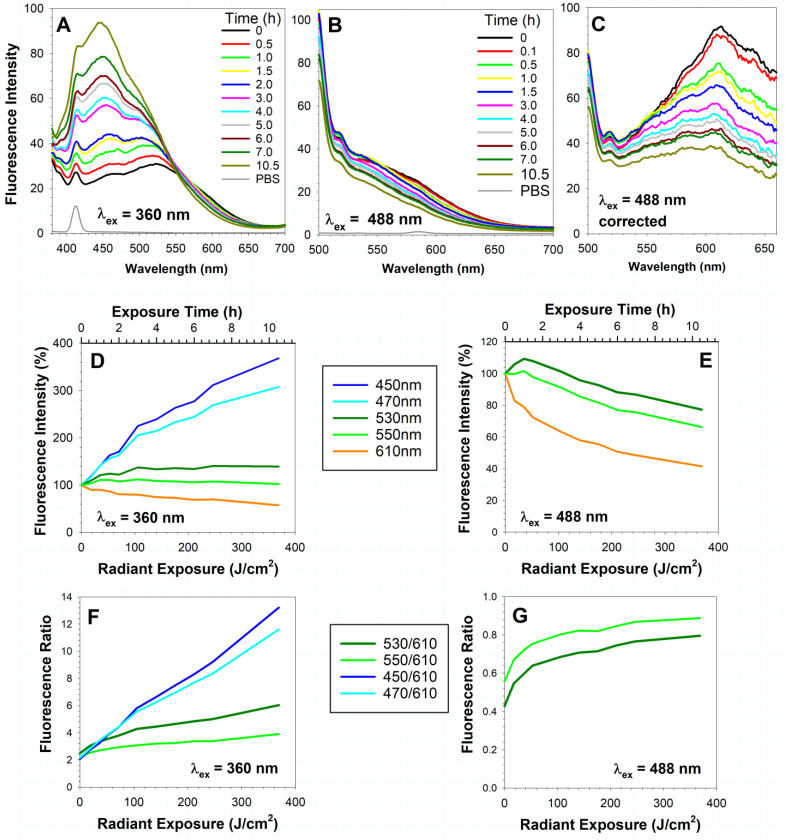
Irradiation of lipofuscin with visible light results in a decrease in long-wavelength fluorescence and an increase in short-wavelength fluorescence. Representative fluorescence emission spectra of lipofuscin granules suspended in PBS before and after indicated time of irradiation with visible light (9.76 mW/cm^2^) upon excitation with 360 nm (**A**) and 488 nm (**B**) light. (**C**) Fluorescence emission spectra from (**B)**, corrected for the spectral changes in sensitivity of the detector, demonstrate that the emission maximum corresponds to the emission maximum of RPE lipofuscin, at about 600 nm. Kinetics of changes of fluorescence emission intensity monitored at indicated wavelengths, normalized to the value before irradiation and expressed as percentage (**D**,**E**) and changes in fluorescence intensity ratios at indicated wavelengths (**F**,**G**) during irradiation with 9.76 mW/cm^2^ visible light. Fluorescence was induced by excitation with 360 nm (**D**,**F**) or 488 nm (**E**,**G**) light. The ratios of fluorescence intensities were calculated based on uncorrected spectra in the case of excitation at 360 nm and on corrected spectra in the case of excitation at 488 nm. Fluorescence of the solvent (PBS) was recorded to show its Raman emission peaks.

**Figure 2 ijms-23-00922-f002:**
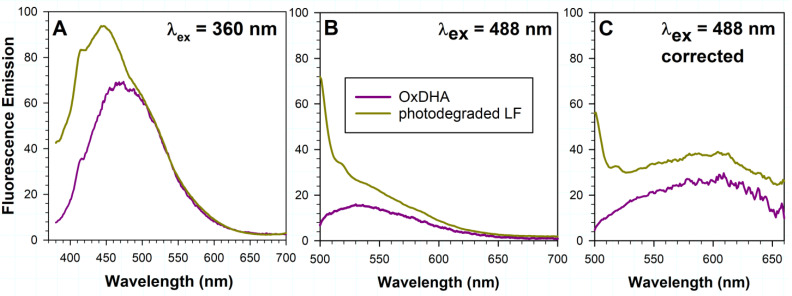
Fluorescence emission of oxidized docosahexaenoate (OxDHA) may contribute to the emission of photodegraded lipofuscin. Representative fluorescence emission spectra of oxDHA and photodegraded lipofuscin (LF) granules obtained by excitation with 360 nm (**A**) and 488 nm (**B**) light. (**C**) Fluorescence emission spectra from (**B**) corrected for the spectral changes in sensitivity of the detector. Prior to fluorescence measurement, lipofuscin was photodegraded by irradiation for 10.5 h with 9.76 mW/cm^2^ visible light providing radiant exposure of 369 J/cm^2^. OxDHA was obtained by auto-oxidation of phosphatidylcholine with two DHA acyl chains. The concentration of OxDHA used for fluorescence measurements corresponds to 4.23 mg/mL of the non-oxidized phospholipid.

**Figure 3 ijms-23-00922-f003:**
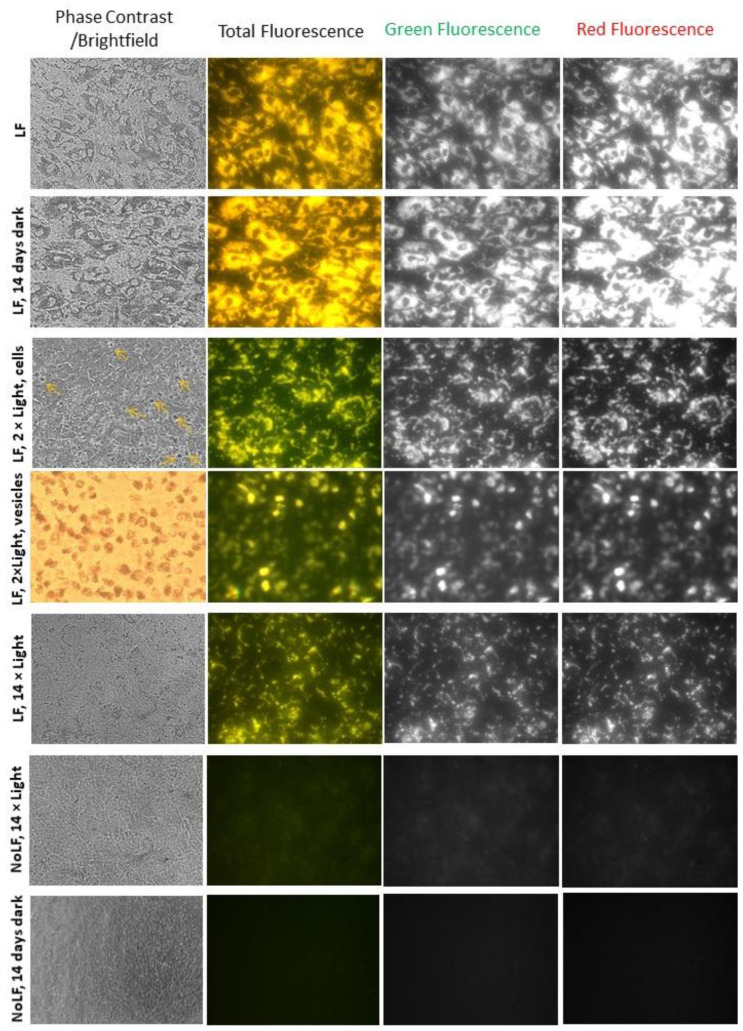
Exposure of lipofuscin-laden ARPE-19 cells to 9.76 mJ/cm^2^ visible light results in exocytosis of packeted lipofuscin granules and spectral changes in lipofuscin fluorescence. Representative images from phase-contrast/bright-field and fluorescence microscopy of ARPE-19 cells supplemented with lipofuscin granules 3 times per week over a period of 32 days (13 times in total) and imaged on day 32 (LF); after additional 14 days in dark with daily media change and incubation for 45 min with PBS (LF, 14 days dark); after additional 3 days which included two 45 min exposures to visible light and images were focused either on cells or vesicles floating above them (LF, 2 × Light, cells; LF, 2 × Light, vesicles). The yellow arrows in the phase-contrast image with 100× magnification point to the packeted lipofuscin floating above the cell monolayer, and images on the right show fluorescence from the cell monolayer (LF, 2 × Light, cells). The row below shows images focused on the packeted lipofuscin in bright-field or fluorescence mode (LF, 2 × Light, vesicles); after 15 days with 14 daily exposures to visible light (LF, 14 × Light). The bottom two rows show cells not supplemented with lipofuscin but treated in the same ways as the supplemented cells with light (NoLF, 14 × Light) or in dark (NoLF, 14 days dark). Fluorescence images were acquired with colour camera capturing the colour of lipofuscin fluorescence as it appears to the human eye (Total Fluorescence). The gain used for collecting fluorescence images was two-fold smaller for lipofuscin-laden cells not exposed to light (LF; LF, 14 days dark) than for all other cells. Unless stated otherwise, magnification was 400×. The images were opened in ImageJ as an RGB stack to quantify the green (Green Fluorescence) and red (Red Fluorescence) fluorescence.

**Figure 4 ijms-23-00922-f004:**
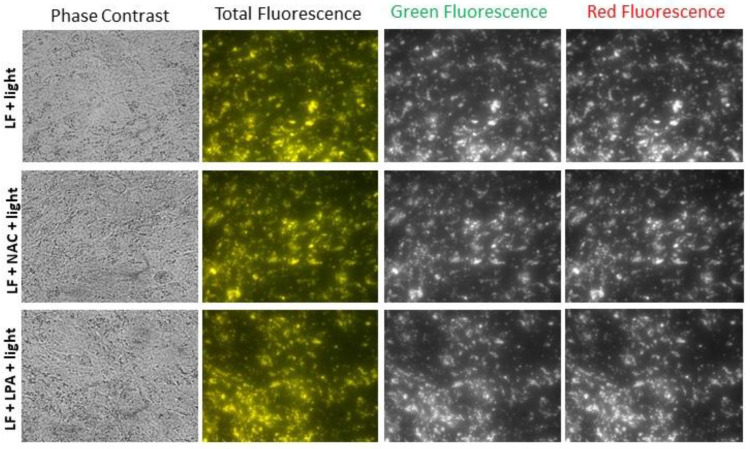
Supplementation with antioxidants of lipofuscin-laden ARPE-19 cells during 14 daily exposures to 9.76 mJ/cm^2^ visible light does not affect loss of lipofuscin granules or spectral changes in fluorescence. Representative images from phase-contrast and fluorescence microscopy of lipofuscin-laden ARPE-19 cells after 14 daily exposures to visible light in the absence (LF + 14 × Light) and presence of N-acetyl cysteine (NAC) or lipoic acid (LPA). Other experimental conditions as in Figure 3.

**Figure 5 ijms-23-00922-f005:**
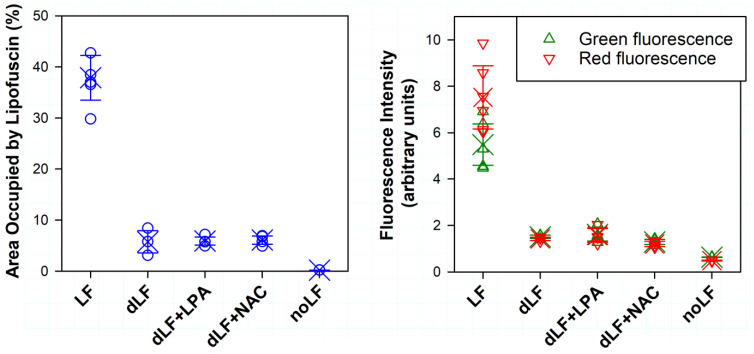
Quantification of area occupied by lipofuscin in phase-contrast images (left panel) and intensity of green and red fluorescence (right panel) from experiments for which the representative images are shown in Figure 3 and Figure 4. LF: lipofuscin-laden cells maintained for additional 15 days with 14 daily media changes and 45 min incubations with PBS; dLF: lipofuscin-laden cells maintained for additional 15 days with 14 daily media changes and 45 min exposures to 9.76 mJ/cm^2^ visible light; dLF+LPA: as dLF but exposures to light were in the presence of 0.1 mM lipoic acid; dLF+NAC: as dLF but exposures to light were in the presence of 0.2 mM N-acetyl cysteine; noLF: as dLF but cells were subjected to sham treatment instead of supplementation with lipofuscin and exposure to light. The circles and triangles represent data from different experiments; the crosses represent the means; the error bars represent SDs.

**Figure 6 ijms-23-00922-f006:**
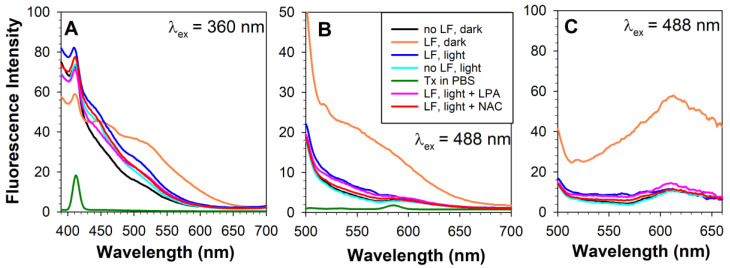
Representative fluorescence emission spectra from ARPE-19 cells solubilized in Triton X-100. Cells were treated as in Figure 3 and Figure 4. Fluorescence was induced by photoexcitation with 360 nm (**A**) or 488 nm (**B**) light. (**C**) Fluorescence emission spectra from (**B**), corrected for the spectral changes in sensitivity of the detector. Confluent cells were supplemented with lipofuscin (LF) or sham-treated (no LF) for 32 days followed by 15 days with 14 daily media changes and 45 min incubations with PBS in dark (dark) or 14 daily media changes and 45 min exposures to 9.76 mJ/cm^2^ visible light (light). Cells in selected wells were exposed to light in the presence of 0.1 mM lipoic acid (LPA) or 0.2 mM N-acetyl cysteine (NAC). Green line shows Raman emission peak from solvent, 1% Triton X-100 in PBS (Tx in PBS).

**Figure 7 ijms-23-00922-f007:**
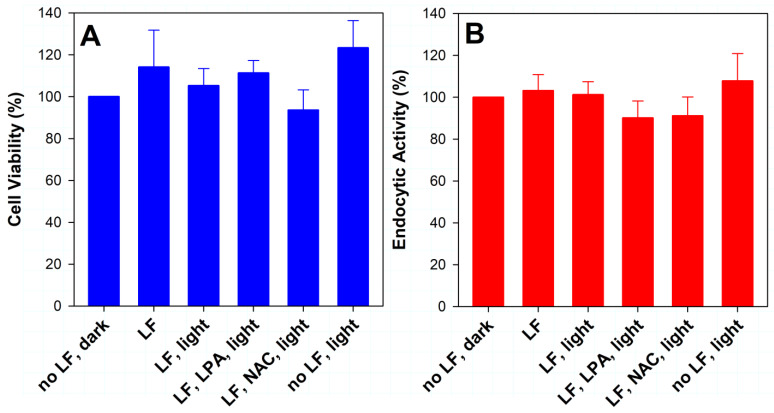
Lipofuscin and/or fractionated exposure to 9.76 mW/cm^2^ visible light does not affect viability or endocytic activity of ARPE-19 cells. (**A**) Cell viability measured by MTT assay. (**B**) Endocytic activity was measured by neutral red assay. Confluent cells were supplemented with lipofuscin (LF) or sham-treated (no LF) for 32 days followed by 15 days with 14 daily media changes and 45 min incubations with PBS in dark (dark) or 14 daily media changes and 45 min exposures to 9.76 mJ/cm^2^ visible light (light). Cells in selected wells were exposed to light in the presence of 0.1 mM lipoic acid (LPA) or 0.2 mM N-acetyl cysteine (NAC). The assays were performed a day after the 14th exposure to light/dark. The absorbance values were normalized to the value obtained in cells not supplemented with lipofuscin and maintained in dark. The height of the bar indicates the mean of three experiments, while the error bar indicates SD. One-way analysis of variance (ANOVA) showed no significant differences between any groups.

**Figure 8 ijms-23-00922-f008:**
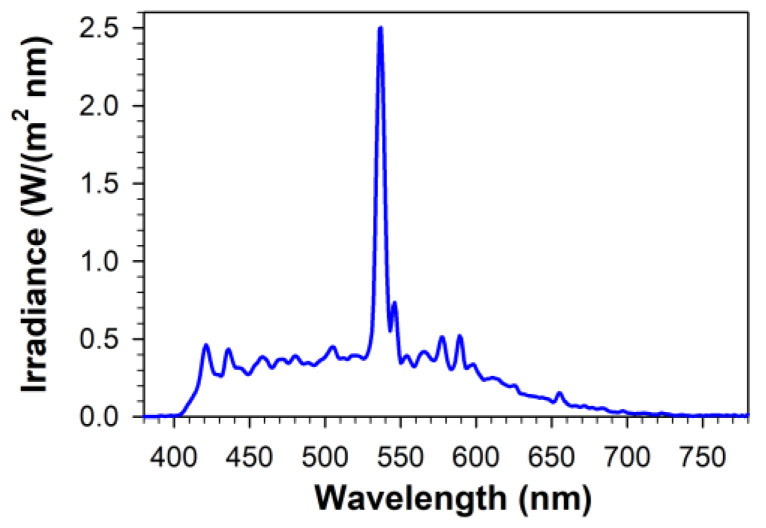
Irradiance spectrum of light used for photodegradation of lipofuscin and exposure to light of ARPE-19 cells.

## Data Availability

The data presented in the manuscript are available upon request.

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
