# Peer review of "Photodegradation of Lipofuscin in Suspension and in ARPE-19 Cells and the Similarity of Fluorescence of the Photodegradation Product with Oxidized Docosahexaenoate"

_ijms, 2022, doi:10.3390/ijms23020922_

Round 1

Reviewer 1 Report

This is a study demonstrating that fluorescence emission spectra and the quantitative evaluation of long-wavelength fluorescence could serve as a monitoring tool for lipofuscin oxidation and hence for screening the retina regarding early AMD changes. It is a research that brings into focus a possible mechanism for AMD pathogenesis and therefore interesting both for researchers and clinical ophthalmologists. Minor revision of English is recommended. 

Author Response

We thank the Reviewer 1 for their critical reading and positive comments on the manuscript. As recommended, we have corrected the typographical errors and amended some sentences to improve their clarity, what can be seen in the manuscript with Track Changes.

Reviewer 2 Report

The manuscript presented by Różanowska and Różanowski in the present form presents some criticalities among which the most important are the confusion in the various sections, for example in the introductory part technical details are inserted that would be more useful in the part of the methods. In addition, the images shown in Figure 3 and Figure 4 are of such low quality that it is not possible to recognize the cell morphology and the total fluorescence panel has a lot of beckground. Also, I would remove the yellow lettering from the field shown as it impedes the view of the image.
The images should be replaced and the sections of the manuscript well separated and organized.

Author Response

We thank the Reviewer 2 for the constructive criticism. We have removed some experimental details from the end of the Introduction where we summarised our approach to accomplish the aims. The images lose some detail when saved in PowerPoint in any format accepted by the publisher, which indeed affect the quality of images especially those from the phase contrast microscopy. Therefore, we included the images in PowerPoint as supplementary files. Images were collected by fluorescence microscopy, not confocal microscopy. What the Reviewer 2 considers as a background fluorescence is the fluorescence coming from the areas outside the focal plane. As requested, the yellow labels were removed from the images and the labels were moved to the left-hand side of each row. We have added to the manuscript the Conclusions section.

Round 2

Reviewer 2 Report

The authors have improved the manuscript so it can be considered for publication.